# Factors Affecting Obesity in Urban and Rural Adolescents: Demographic, Socioeconomic Characteristics, Health Behavior and Health Education

**DOI:** 10.3390/ijerph18052405

**Published:** 2021-03-01

**Authors:** Gyu-Young Lee, Youn-Joo Um

**Affiliations:** 1Red Cross College of Nursing, Chung-Ang University, 84 Heukseok-ro, Dongjak-gu, Seoul 06974, Korea; queyoung@cau.ac.kr; 2Graduate School of Nursing, Chung-Ang University, 84 Heukseok-ro, Dongjak-gu, Seoul 06974, Korea

**Keywords:** urban health, rural health, adolescent, health equity, obesity

## Abstract

In this study, we aimed to analyze the demographic factors, socioeconomic factors, health behavior factors, and health education of urban and rural adolescents and their impact on obesity in rural and urban youths. We analyzed data from 60,040 students from 800 schools using the 14th Youth Health Behavior Web-based Survey data for 2018. The mean age of the participants was 15.01 ± 1.77; 30,463 (50.7%) males and 29,577 (49.3%) females. The data were analyzed using a chi-square test and multinominal logistic regression analysis. Rural adolescents had significantly lower parental income, parental education, and academic performance levels. Urban adolescents tended to have a higher rate of breakfast fasting, ate more fast-food, exercised less, had a higher rate of sleep dissatisfaction, and had significantly higher current alcohol intake. Rural adolescents reported more stress and obesity levels than their urban counterparts. Factors influencing the obesity of urban adolescents were grade level, father’s education, mother’s education, academic performance level, fast-food, exercise, current alcohol intake, and sleep satisfaction. Factors influencing the obesity of rural adolescents were parents’ income level, academic performance level, exercise, and sleep satisfaction. To effectively prevent and manage adolescent obesity, which is an indicator of health equity among adolescents, an effort must be made to improve health education and the local environment in both urban and rural areas.

## 1. Introduction

Socioeconomic status has a profound impact on an individual’s health and can be a root cause of disease [1]. Poor socioeconomic status is associated with poor health, while high socioeconomic statuses are associated with relatively good health [2]. These socioeconomic statuses lead to health disparities and act as health determinants, negatively affecting health equity [3]. Health equity exists when population groups (i.e., based on social, demographic, economic, and regional factors) have fair and equal access to healthcare and there are no inevitable health disparities between groups [4]. Previous studies have shown that economic, environmental, and social inequality between population groups affects the risk of disease and the ability to prevent disease [5]. Besides this, personal factors, such as smoking, drinking, nutrition and exercise, and social and community circumstances, contribute to health inequality [5]. As such, the health of individuals and communities is affected simultaneously by a combination of factors, which can lead to health disparities. The World Health Organization has said that determining health inequality and its reasons are essential to achieving health equity [6].

Interest in regional health disparities began in earnest in 1980, with the publication of the Black Report in the UK [7]. In this report, a study on health inequality was conducted, and it was found that health disparities became more severe as the social class went down to simple labor [7]. Since then, various studies on health inequality have been conducted in the US and the UK [7,8]. In Korea in the 2000s, the issue of health inequality between the rich and the poor was brought to light, causing an increase in research on the subject [9]. According to previous studies, it was found that health inequality exists between the metropolitan area and the non-metropolitan area, and between cities and rural areas and that local health status is closely related to regional health inequality [10,11]. Finally, the Ministry of Health and Welfare set health inequality on the agenda in 2005, and set enhancing health equity as a goal of the second National Health Promotion Plan; this effort was continued in the fourth National Health Promotion Plan of 2020 [12]. The school population must not be overlooked to achieve the health promotion goal of Korea, which is the improvement of health equity, and there are insufficient policies and studies to alleviate health inequality among adolescents [13]. It is necessary to examine the factors of health inequality caused by regional gaps such as urban and rural areas.

According to Article 23, Paragraph 1 of the Enforcement Decree of the School Health Act of Korea, each school is supposed to have one health teacher. However, the standards for the placement of health professionals in schools are not observed [14]. In rural areas where there are many small schools, the placement rate of health teachers is low and there is a high possibility that little or no health education will be provided, which may result in differences in the formation of healthy lifestyles between urban and rural areas [14]. A study on health behavior according to the absence or presence of school health education showed that students in small and medium-sized cities showed significantly lower rates of health education than students in large cities [15]. Further, it was reported that students who did not receive school health education had a high rate of skipping breakfast, a low rate of physical activity, and a higher rate of smoking and drinking [15]. According to the 2018 data of the student health test, which measures the health status of students, including height and weight, obesity was found among 15% and 18% of elementary school students in urban and rural areas, respectively, and among 16.0% and 17.3% of middle school students in urban and rural areas, respectively [16].

Obesity in adolescence results in five times the risk of obesity in adulthood and significantly increases the risk of major chronic diseases; hence, prevention and management are essential [17]. While many studies about healthy equity in Korea have been conducted, few studies have comprehensively identified the income gap, health disparity, and health education experience gap based on living in urban or rural areas [4,18,19,20,21]. Since adolescents have much time to receive education at school, it is necessary to diversify obesity prevention education using this. Such efforts to prevent obesity in youth can be expected to be more effective through efforts through policy implementation. Therefore, research on basic data identifying regional differences and health disparities by examining the health behavior, health education experience, and degree of obesity of urban and rural adolescents, is needed to establish a more comprehensive health promotion policy.

Therefore, this study examined the demographic characteristics, socioeconomic characteristics, health behavior, health education experience, and differences between urban and rural adolescents in weight status, and determined the factors affecting obesity in urban and rural adolescents. This study provides basic data for improving the health equity of adolescents by region and establishing customized student health promotion policies for each region. The hypotheses of this study are as follows:

**Hypothesis** **1** **(H1).***There are differences in the demographic characteristics and the socioeconomic characteristics of urban and rural adolescents*.

**Hypothesis** **2** **(H2)**.*There are differences in the health behaviors and health education experiences of urban and rural adolescents*.

**Hypothesis** **3** **(H3)**.*There are differences in the factors that affect obesity in urban and rural adolescents*.

## 2. Materials and Methods

### 2.1. Data Collection

This cross-sectional, descriptive study analyzed data from the 14th Youth Health Behavior Survey conducted nationwide in Korea in 2018 by the Ministry of Education, Science and Technology, the Ministry of Health and Welfare, and the Centers for Disease Control and Prevention. The self-reported online survey was conducted in school computer labs from 1 June to 17 July 2017 [22]. The survey took about 45–50 min per respondent, and the survey content is the same in Appendix A

For the raw data, the target population was defined as national middle and high school students as of April 2017. High schools were divided into general high schools and specialized high schools. The sampling process used in this study is a stratified clustering extraction method. The stratified cluster extraction method is a method of first stratifying, then extracting the primary extraction unit as a cluster, and performing a total survey or sample survey of the extracted clustering. The primary extraction unit was school and was selected as a permanent random number extraction method. For the second extraction, one class was randomly selected for each grade from the selected sample schools, and all the class students were surveyed. Students with long-term absences, disabilities, and text-reading disabilities were excluded. The response ratio was calculated by using the response rate for each grade of the sample school and was calculated as the ratio of the participants who participated in the survey among the number of participants by a grade of the sample school. The weighted post-correction rate was calculated so that the sum of the weights by gender, school level (middle school, high school), and grade in the region was equal to the number of middle and high school students nationwide as of April 2018. As a result, 30,229 middle school respondents and 29,811 high school respondents represented 1,323,788 middle school students nationwide and 1,526,330 high school students nationwide as of April 2018. The participants were students from the first year of middle school to the third year of high school. In all, 62,823 students from 800 schools were asked to participate, of which 60,040 students participated in the survey. Thus, the participation rate was 95.6%, based on the number of students. The raw data is the same as the data of this study.

### 2.2. Instruments

#### 2.2.1. Regions

For the division of urban and rural areas by region, we classified the 79 large cities and 84 small and medium-sized cities as urban, and the 66 counties as rural.

#### 2.2.2. Demographic Characteristics

Demographic characteristics were sex (male, female), grade (1st middle school, 2nd middle school, 3rd middle school, 1st high school, 2nd high school, 3rd high school), school (middle school, high school), and age.

#### 2.2.3. Socioeconomic Characteristics

Based on information from preceding studies [18,19,20,21], the socioeconomic characteristic variables were father’s education, mother’s education, parents’ income level, perceived health status, and academic performance level. Father’s and mother’s education was divided into “≤High school,” “≥University,” and “Unknown.” Parents’ income level was divided into “High,” “Average,” and “Low.” Perceived health status was divided into “Healthy,” “Average,” and “Not healthy.” The academic performance level was divided into “High,” “Average,” and “Low.”

#### 2.2.4. Health Behavior and Health Education

For health behavior variables, we referred to Korea’s Health Plan 2020 Youth Health Living Practice Index [8]. Health behavior variables included breakfast intake, fast-food intake, exercise, sleep satisfaction, level of stress, current alcohol intake, and current smoking.

These seven variables were reclassified [9,23] by referring to prior research following the purpose of the study. Breakfast and fast-food intake were classified as “Yes” or “No” depending on the intake once a week or more. The exercise was classified into three categories: “Not done,” “1–3 times a week,” and “4 days a week or more.” The level of sleep satisfaction was classified as “‘Enough,” “Average,” or “Not enough” based on the answer to the question “Do you think the time spent sleeping in the last 7 days was enough to recover from fatigue?” The level of stress was classified as “A lot,” “Average,” or “None” based on the answer to the question “How much stress do you usually feel?” Current alcohol intake was classified as “Yes” or “No” depending on whether one or more drinks were consumed in the last 30 days. Current smoking was classified as “Yes” or “No” based on the answer to the question “Have you smoked one day or more in the last 30 days?”

For the health education variables, alcohol prevention education was classified as “Yes” or “No” based on the answer to the question “Have you ever received alcohol prevention education at school (including during class hours, broadcasting education, or education in the auditorium, etc.)?” and nutrition education was classified as “Yes” or “No” based on the answer to the question “Have you ever received nutrition education at school (including during class hours, broadcasting education, or education in the auditorium, etc.)?”

#### 2.2.5. Obesity

The obesity variable was the dependent variable of this study. It was classified according to weight status. The criteria for weight classification determined by the Korea Centers for Disease Control and Prevention [24] and the Centers for Disease Control and Prevention [25] are as follows. Depending on the sex and age of the child/adolescents, underweight is classified as Less than the 5th percentile, normal is 5th percentile to less than the 85th percentile, overweight is 85th to less than the 95th percentile, and obesity is classified as Equal to or greater than the 95th percentile. In this study, less than the 5th percentile of BMI was classified as the “underweight” group, the 5th percentile to less than the 85th percentile were classified as the “normal” group, and the 85th percentile or more were classified as “overweight or obese” group [26]. The body measurement was based on the recent measurement of height and weight reported by the respondent to the first decimal place, and the Body Mass Index (BMI) was calculated by the Weight/Height ^2^ (kg/m^2^) formula.

### 2.3. Data Analysis

In this study, out of 62,323 students in the sample school, students who answered incorrectly or who did not participate were excluded, and a total of 60,040 students were analyzed. None of the variables in this study had missing values and a normal distribution was assumed. The data were analyzed using SPSS WIN version 25.0 (SPSS Inc., Chicago, IL, USA), and the significance level of the statistics was set to *p* < 0.05. To compare the demographic characteristics, the socioeconomic characteristics, health behavior, and health education of urban and rural adolescents, frequency analysis and χ^2^-test were performed. Besides this, Pearson correlation analysis was performed on all variables to analyze the correlation between variables. In this study, after analyzing differences in demographic characteristics, socioeconomic characteristics, health behavior, and health education, a multinominal logistic regression analysis was performed with only significant *p* < 0.05 data. Multinominal logistic regression analysis was performed to identify the factors affecting obesity in urban and rural areas, and the significance level was set to *p* < 0.05. Furthermore, we performed a multinominal logistic regression analysis to determine the effect of each variable with all variables calibrated for overweight.

### 2.4. Ethical Considerations

This study was a government-approved statistical survey (approval number 11757) based on Article 19 of the National Health Promotion Act. The original data were downloaded from the website of the Korea Centers for Disease Control and Prevention and processed on 1 April 2019, after approval was obtained for the use of the raw data from the Korea Centers for Disease Control and Prevention by the Regulations on the Disclosure and Use of Online Source Data for Youth Health Behavior [22]. The contents and methods of this study were approved by the University Bioethics Committee (IRB NO: 1041078-201902-HRSB-064-01C).

## 3. Results

### 3.1. Comparison of Demographic Characteristics of Urban and Rural Adolescents

Table 1 shows the demographic characteristics of the 60,040 participants, of which 30,463 were males (50.7%) and 29,577 were females (49.3%). There were 55,514 (92.5%) students from urban areas and 4526 (7.5%) from rural areas. In rural areas, the percentage of high school students was lower (45.7%) than in urban (50.0%). The average age was 15.02 ± 1.77 in cities and 14.91 ± 1.75 in rural areas.

### 3.2. Correlation Analysis of Research Variables

Table 2 shows the results of an analysis of the correlation of each variable in urban and rural areas.

### 3.3. Comparison of Socioeconomic Characteristics of Urban and Rural Adolescents

Table 3 shows the socioeconomic characteristics of the subjects. In urban areas, fathers (χ^2^ = 675.59, *p <* 0.001) and mothers (χ^2^ = 486.85, *p* < 0.001) tended to have more education. Low parental income was reported by a larger proportion of students in rural areas than urban areas, and high parental income was reported by a larger proportion of students in urban areas than rural areas (χ^2^ = 33.69, *p* < 0.001). Low academic performance levels were reported by a slightly larger proportion of students in rural areas than urban areas, and high academic performance was reported by a slightly larger proportion of students in urban areas than the rural (χ^2^ = 8.67, *p* = 0.010).

### 3.4. Comparison of Health Behavior and Health Education of Urban and Rural Adolescents

Table 4 shows a comparison of the health behaviors and health educations of urban and rural adolescents. Adolescents in urban areas said that they did not have breakfast (χ^2^ = 6.65, *p* = 0.010) and consumed fast-food (χ^2^ = 19.76, *p* < 0.001) more often than those in rural areas. Besides this, adolescents in urban areas reported no exercise and insufficient sleep more often than adolescents in rural areas (χ^2^ = 12.78, *p* = 0.002). Adolescents in urban areas reported a higher rate of current alcohol intake than those in rural areas (χ^2^ = 23.20, *p* < 0.001), and a larger percentage of adolescents in rural areas reported overweight or obesity compared to those in urban areas (χ^2^ = 21.33, *p* < 0.001). 

### 3.5. Factors Affecting Obesity of Adolescents in Urban and Rural Adolescents

To identify the factors influencing obesity in adolescents in urban and rural areas, independent variables that showed statistically significant differences (*p* < 0.05) were analyzed. These independent variables included the school grade, father’s education, mother’s education, parents’ income level, academic performance level, breakfast, fast-food, exercise, current alcohol intake, sleep satisfaction, and stress level. To accurately analyze the factors influencing obesity in urban and rural adolescents, a multinomial logistic regression analysis was conducted in three groups: underweight, normal, and overweight or obesity. The results are shown in Table 5. Factors influencing overweight or obesity in urban adolescents were the school grade, father’s education, mother’s education, academic performance level, fast-food, exercise, current alcohol intake, and sleep satisfaction. In the case of urban adolescents, the risk of overweight or obesity level for middle school students was lower compared to that for high school students (OR = 0.71, *p* < 0.001), and the risk of overweight or obesity was higher when father’s education (OR = 1.16, *p* < 0.001) or mother’s education (OR = 1.09, *p* = 0.005) was lower than “≥University”. The risk of overweight or obesity was higher for urban adolescents with a “low” academic performance level (OR = 1.22, *p* < 0.001). Urban adolescents who responded affirmatively to consuming fast-food had a higher risk of overweight or obesity than those who responded negatively (OR = 1.15, *p* < 0.001). Urban adolescents who responded that they exercised had a lower likelihood of overweight or obesity compared to those who reported that they never exercise (OR = 0.82, *p* < 0.001). Further, urban adolescents who reported alcohol intake had a higher risk of overweight or obesity than those who did not (OR = 1.10, *p* = 0.001). Urban adolescents who reported that their sleep satisfaction was “Not enough” had a higher risk of obesity than those who reported “Enough” or “Average” (OR = 1.23, *p* < 0.001). Factors influencing underweight in urban adolescents were the school grade, current alcohol intake, and sleep satisfaction. In the case of urban adolescents, the risk of underweight level for middle school students was lower compared to that for high school students (OR = 2.27, *p* < 0.001). Further, urban adolescents who reported alcohol intake had a higher risk of underweight than those who did not (OR = 0.86, *p* < 0.001). Urban adolescents who reported that their sleep satisfaction was “Not enough” had a higher risk of underweight than those who reported “Enough” or “Average” (OR = 1.08, *p* < 0.001).

Factors influencing overweight or obesity in rural adolescents were the parents’ income level, academic performance level, and sleep satisfaction. In the case of rural adolescents, the risk of overweight or obesity was lower for adolescents whose parents’ income level was reported as “Low” compared to adolescents whose parents’ income level was “Average” or “High” (OR = 0.60, *p* = 0.011). Besides this, rural adolescents who reported “Low” academic performance levels had a higher risk of overweight or obesity than those who reported “Average” or “High” academic performance levels (OR = 1.32, *p* < 0.001). Rural adolescents who reported that they exercised had a lower risk for overweight or obesity compared to those who reported that they had never exercised (OR = 0.77, *p* < 0.001). Rural adolescents who reported that their sleep satisfaction was “Not enough” had a higher risk of obesity than those who reported “Enough” or “Average” (OR = 1.15, *p* = 0.013). Factors influencing underweight in rural adolescents was the school grade. In the case of rural adolescents, the risk of underweight level for middle school students was lower compared to that for high school students (OR = 2.36, *p* < 0.001). 

## 4. Discussion

This study analyzed secondary data from the 14th Youth Health Behavior Survey [22] conducted in 2018 to determine whether differences in demographic characteristics, the socioeconomic characteristics, health behaviors, and health education experience among urban and rural adolescents impact obesity. The purpose of this study was to prepare basic data that can be used to improve the health equity of urban and rural adolescents and develop customized health promotion policies for each region.

The results of this study are as follows. First, there was a difference in socioeconomic characteristics of urban and rural areas. In rural, the number of students in each grade reduced as the grade increased. In rural areas, particularly, the number of students in the first year of high school was significantly lower than the number of students in the third year of middle school. Further, the results show that the father’s education, mother’s education, parents’ income level, and academic performance level in rural areas were significantly lower than in urban areas.

Second, there were differences in health behavior and health education experiences in urban and rural areas. According to this study, urban adolescents are more likely to skip breakfast and consume fast-food, have a lower exercise and sleep satisfaction rate, and have a significantly higher current alcohol intake rate, making them more vulnerable to health issues. The results of this study show that the obesity rate for rural adolescents is high, and the current alcohol intake rate and current smoking rate are significantly higher for urban adolescents.

Third, there was a difference in the factors affecting the obesity of adolescents in urban and rural areas. Factors influencing the obesity of urban adolescents were grade level, father’s education, mother’s education, academic performance level, fast-food, exercise, current alcohol intake, and sleep satisfaction. Factors influencing the obesity of rural adolescents were parents’ income level, academic performance level, exercise, and sleep satisfaction. Factors affecting underweight in urban adolescents were the school grade, current alcohol intake, and sleep satisfaction, and the factor affecting underweight in rural adolescents was school grades.

Comprehensively, the number of students decreased as they went to high school because rural considered that the educational environment for college admission was relatively poorer than in cities [27,28]. In many cases, only those students with difficult family circumstances, such as those living with grandparents or in single-parent families, remain in the rural areas, which can further increase the educational gap between urban and rural areas [27]. This suggests that more intensive socioeconomic and educational support is needed for rural areas. These results are like the results in the existing literature, which notes that the socioeconomic status of a family determines the level of academic achievement of the children [28]. The Seoul Institute’s Policy Task Research Report also found that the BMI of adolescents was negatively correlated with parents’ income and education level [29]. In addition, there is a prior study that rural adolescents showed physical symptoms due to psychological stress [30]. Rural students are more stressed [27] due to poor educational environments, but further research is needed to analyze regional stress differences through objective stress measures. On the other hand, with regard to urban, it is in line with the study that the physical strength of urban students is weaker than that of rural students, and urban students’ bedtime is significantly slower than rural students [31]. Urban adolescents may not have enough time to focus on their health because they spend more time on private education [27] and live in environments where fast-food is more widely distributed. Therefore, policies are needed to create a local environment that can protect health in the daily lives of urban adolescents and reduce access to fast-food. Therefore, school health education should be carried out to improve exercise habits and sleep quality, along with efforts to implement healthy eating habits, reduce fast-food consumption and introduce customized health promotion policies for urban adolescents. On the other hand, there was no significant difference between urban and rural areas in perceptions of subjective health. It was used as a control variable, but this is consistent with a previous study comparing the subjective health status of adolescents residing in the metropolitan area, metropolitan cities, and other areas [32]. As independence increases according to the characteristics of adolescence, the importance of relative evaluation through friends around parents is gradually increasing rather than the socioeconomic status of parents [33]. This seems to be because adolescents recognize their health status through comparison with their surroundings [34].

Adolescents with higher parental education or healthy eating habits were found to have a lower probability of obesity. In this study, long-term absenteeism students, students with disabilities, and students with text reading disabilities were excluded. However, as for the obesity rate of adolescents with mental disorders, it was found that the higher the socioeconomic characteristics of the parents and the better the diet and exercise were, the less obese the students [35]. Therefore, in summary, socioeconomic status has a significant influence on the obesity of adolescents, and it is necessary to examine this in more depth in future studies.

Therefore, to reduce socioeconomic inequality and increase health equity, the role of local community government, as well as individual health behavior and health education, is important. It was found that local community governments should strengthen their socioeconomic capacity and reduce the risk of obesity in the region by improving the physical environment such as parks. Urban areas need to emphasize policies that can restrict access to harmful foods, such as fast-food, and that give attention to mental health, thereby improving the quality of sleep and creating a healthy city that helps adolescents who are too busy to relax. On the other hand, since rural areas lack infrastructure for health, there is a need to implement local community health programs, health services, and health education. Further, there is a need to prepare policies to improve socioeconomic capacity, such as preparing local jobs and revitalizing the economy. Adolescent obesity is affected by the physical environment, so the neighborhood environment is important, and it has been shown that the relationship with the socioeconomic capacity of individuals and society is significant, so policy establishment in this direction is necessary. Besides this, local governments should take the direction of providing public services in creating and supporting projects by actively inducing private cooperation between public institutions and administrative agencies. It is not easy to change the region by showing leadership only in the public sector. It is important to create an atmosphere and an environment that empowers the private sector and enables common leadership. In addition to universal services for discovering blind spots and improving the level of inequality across the population, customized services in proportion to the needs of the vulnerable are added. The program can be composed of a short-term program to improve health behavior, a mid-term program to establish a community network, and a long-term program dealing with socioeconomic factors.

This study also has some limitations. First, we used large-scale, nationwide data, but the original survey did not measure all health-related characteristics and obesity-related temporal changes. In future research, it is proposed to compare obesity in urban and rural areas according to temporal changes through longitudinal studies. Second, due to its cross-sectional study design, there is a limit to the study in terms of causality. In the future, a prospective study using objectified data that can consider the causal relationship related to obesity should be conducted based on the results of this study. Third, since the study was conducted using a self-reported questionnaire, there may be self-report bias in the results. Therefore, it seems necessary to apply an in-depth interview or observation method to confirm the socioeconomic, health behavior, and health education characteristics of urban and rural adolescents for future studies. Since height and weight were collected by self-report, accuracy may be inferior. In future studies, it will be necessary to measure more accurately using more formal records. In addition, when collecting data on health behavior (breakfast, fast-food, sleep quality, stress level, etc.), it may be insufficient to include only one question. Therefore, it is necessary to understand and analyze these in more detail as multiple-choice items in future research. Besides this, this study compared underweight and normal, overweight and obesity, but it is necessary to subdivide and analyze underweight, normal, overweight, and obesity. Further, there may be a bias in the research results due to large differences in sample sizes of urban and rural adolescents. To improve accuracy, it is necessary to analyze the sample size in urban and rural areas by adjusting similarly in future studies. Finally, although it has limitations in that it does not include students who did not come to school, the representative of the data is very high because the national health statistics data are used, and, significantly, the basic data for the health equity of adolescents in Korea were prepared.

## 5. Conclusions

This study identifies the effects of demographic, socioeconomic, health behavior, and health education factors on obesity, which is widely used as a health equity indicator, in rural and urban areas. It is necessary to take note of health disparities between urban and rural adolescents and ensure access to necessary resources and improvements in each region through the influencing factors identified in this study, which should be utilized in the development of adolescent health promotion programs in the future. In addition, it is necessary to identify policy measures to improve the health inequality factors of adolescents in the community and to study arbitration strategies that take community differences into account.

## Figures and Tables

**Table 1 ijerph-18-02405-t001:** Participant demographic characteristics.

Variables	Categories	Total (*n* = 60,040)	Urban (*n* = 55,514)	Rural (*n* = 4526)	χ^2^ or t (*p*)
		*n*(%)/M ± SD	*n*(%)/M ± SD	*n*(%)/M ± SD	
Sex	Male	30,463 (50.7)	28,170 (50.7)	2293 (50.7)	0.01 (0.461)
	Female	29,577 (49.3)	27,344 (49.3)	2233 (49.3)	
Grade	Middle school (Year 1)	9847 (16.4)	9064 (16.3)	783 (17.3)	
	Middle school (Year 2)	10,092 (16.9)	9277 (16.7)	815 (18.0)	33.03 (<0.001)
	Middle school (Year 3)	10,290 (17.1)	9430 (17.0)	860 (19.0)	
	High school (Year 1)	9260 (15.4)	8620 (15.5)	640 (14.1)	
	High school (Year 2)	10,039 (16.7)	9322 (16.8)	717 (15.8)	
	High school (Year 3)	10,512 (17.5)	9801 (17.7)	711 (15.7)	
School	Middle school	30,229 (50.3)	27,771 (50.0)	2458 (54.3)	30.54 (<0.001)
	High school	29,811 (49.7)	27,743 (50.0)	2068 (45.7)	
Age (years)	Mean	15.01 ± 1.77	15.02 ± 1.77	14.91 ± 1.75	4.22 (<0.001)
	Min	12	12	12	0.000 (1.0)
	Max	18	18	18	0.000 (1.0)

Data are presented as the number and percentage of respondents per variable. *p*-values were obtained from the χ^2^ test to investigate the difference between urban and rural. Total is the total number of participants who do not differentiate between urban and rural areas.

**Table 2 ijerph-18-02405-t002:** Correlation of variables in urban and rural areas.

	Rural	Sex (1)	School (2)	Age (3)	Father’s Education (4)	Mother’s Education (5)	Parents’ Income Level (6)	Perceived Health Status (7)	Academic Performance Level (8)	Breakfast (9)	Fast-Food (10)	Exercise (11)	Sleep Satisfaction (12)	Level of Stress (13)	Current Alcohol (14)	Current Smoke (15)	Alcohol Prevention Education (16)	Nutrition Education (17)	BMI (18)
Urban																			
(1)		1	0.012	0.014	0.020	0.044 *	−0.038 *	−0.116 **	−0.017	0.013	−0.002	0.268 **	−0.159 **	0.232 **	0.055 **	0.107 **	−0.059 **	−0.080 **	−0.138 **
(2)		−0.002	1	0.839 **	0.094 **	0.150 **	−0.104 **	−0.075 **	−0.092 **	0.024	0.057 **	0.151 **	−0.176 **	0.103**	−0.101 **	−0.092 **	−0.053 **	−0.199 **	0.155 **
(3)		−0.001	0.843 **	1	0.116 **	0.150 **	−0.112 **	−0.085 **	−0.072 **	0.026	0.058 **	0.132 **	−0.173 **	0.103 **	−0.114 **	−0.101 **	−0.065 **	−0.229 **	0.167 **
(4)		0.009	0.075 **	0.090 **	1	0.518 **	−0.124 **	−0.032	−0.174 **	−0.056 **	0.015	0.007	0.005	0.040 *	−0.044 *	−0.006	−0.012	−0.041 *	0.080 **
(5)		0.037 **	0.106 **	0.122 **	0.549 **	1	−0.109 **	−0.048 **	−0.152 **	−0.055 **	0.044 *	0.036 *	−0.031	0.052 **	−0.024	−0.010	−0.045 **	−0.079 **	0.045 *
(6)		−0.063 **	−0.084 **	−0.096 **	−0.129 **	−0.133 **	1	0.104 **	0.133 **	0.024	−0.010	−0.055 **	0.064 **	−0.080 **	−0.010	0.007	0.006	0.048 **	−0.021
(7)		−0.113 **	−0.116 **	−0.129 **	−0.040 **	−0.048 **	0.110 **	1	0.084 **	0.034 *	−0.015	−0.148 **	0.180 **	−0.252 **	0.026	0.024	0.021	0.069 **	−0.060 **
(8)		−0.025 **	−0.110 **	−0.114 **	−0.156 **	−0.149 **	0.171 **	0.105 **	1	0.080 **	−0.032 *	−0.032 *	0.016	−0.061 **	0.016	0.047 **	0.089 **	0.066 **	−0.064 **
(9)		0.004	−0.001	−0.005	−0.052 **	−0.048 **	0.016 **	0.043 **	0.094 **	1	0.004	−0.046 **	0.025	−0.027	−0.006	0.006	0.036 *	0.060 **	0.001
(10)		−0.028 **	0.053 **	0.065 **	0.010 *	0.013 **	−0.013 **	−0.008 *	−0.024 **	0.003	1	−0.010	−0.047 **	0.017	0.011	−0.015	0.007	0.015	−0.004
(11)		0.284 **	0.120 **	0.120 **	0.012 *	0.025 **	−0.059 **	−0.161 **	−0.026 **	−0.037 **	−0.004	1	−0.063 **	0.099 **	0.033 *	0.009	−0.056 **	−0.144 **	−0.031 *
(12)		−0.169 **	−0.222 **	−0.225 **	−0.012 *	−0.036 **	0.076 **	0.199 **	0.040 **	0.038 **	−0.038 **	−0.093 **	1	−0.325 **	0.086 **	0.058 **	0.041 **	0.069 **	0.006
(13)		0.204 **	0.080 **	0.086 **	0.026 **	0.039 **	−0.082 **	−0.266 **	−0.078 **	−0.033 **	0.025 **	0.094 **	−0.315 **	1	−0.029	−0.017	−0.018	−0.063 **	0.030 *
(14)		0.056 **	−0.117 **	−0.137 **	−0.049 **	−0.048 **	0.028 **	0.014 **	0.059 **	0.023 **	−0.020 **	0.019 **	0.074**	−0.037 **	1	0.141 **	0.025	0.022	0.038 *
(15)		0.131 **	−0.087 **	−0.104 **	−0.039 **	−0.041 **	0.017**	0.005	0.066 **	0.030 **	−0.029 **	0.050 **	0.037 **	−0.006	0.140 **	1	0.005	0.034 *	0.030 *
(16)		−0.030 **	−0.128 **	−0.141 **	−0.027 **	−0.029 **	0.014 **	0.036 **	0.079 **	0.027 **	−0.008	−0.070 **	0.062 **	−0.037 **	0.041 **	0.023 **	1	0.293**	0.016
(17)		−0.053 **	−0.214 **	−0.233 **	−0.027 **	−0.034 **	0.037 **	0.085 **	0.080 **	0.035 **	−0.001	−0.096 **	0.113 **	−0.074 **	0.042 **	0.026 **	0.312**	1	−0.050 **
(18)		−0.157 **	0.187 **	0.197 **	0.075 **	0.066 **	−0.028 **	−0.056 **	−0.072 **	−0.015 **	−0.008 *	−0.054 **	−0.016 **	0.030 **	0.057 **	0.049 **	−0.024 **	−0.039 **	1

Pearson correlation analysis was performed, * *p* < 0.05, ** *p* < 0.01, Sex: 0 = male, 1 = female; Father’s education: 1 = Middle school graduated, 2 = Highschool graduated, 3 = University graduated, result value substition = Unknown; Mother’s education: 1 = Middle school graduated, 2 = Highschool graduated, 3 = University graduated, result value substition = Unknown; Parents’ income level: 1 = Low, 2 = Average, 3 = High; Perceived health status: 1 = Low, 2 = Average, 3 = High; Academic performance level: 1 = Low, 2 = Average, 3 = High; Breakfast: 1 = No, 2 = Yes; Fast-food: 1 = No, 2 = Yes; Exercise: 1 = Not Done, 2 = 1~3 days, 3 = 4 ≥ days; Sleep satisfaction: 1 = Not enough, 2 = Average, 3 = Enough; Level of stress: 1= None, 2 = Average, 3 = A lot; Current alcohol intake: 1 = No, 2 = Yes; Current smoke intake: 1 = No, 2 = Yes; Alcohol prevention education: 1 = No, 2 = Yes.

**Table 3 ijerph-18-02405-t003:** Socioeconomic characteristics.

Variables	Categories	Total (*n* = 60,040)	Urban (*n* = 55,514)	Rural (*n* = 4526)	χ^2^ (*p*)
		*n* (%)/M ± SD	*n* (%)/M ± SD	*n* (%)/M ± SD	
Father’s education	≤High school	15,260 (25.4)	13,615 (24.5)	1645 (36.4)	675.59 (<0.001)
	≥University	31,236 (52.0)	29,674 (53.5)	1562 (34.5)	
	Unknown	13,544 (22.6)	12,225 (22.0)	1319 (29.1)	
Mother’s education	≤High school	17,864 (29.7)	16,140 (29.1)	1724 (38.1)	486.85 (<0.001)
	≥University	29,581 (49.3)	28,056 (50.5)	1525 (33.7)	
	Unknown	12,595 (21.0)	11,318 (20.4)	1277 (28.2)	
Parents’ income level	High	6526 (10.9)	6118 (11.0)	408 (9.0)	33.69 (<0.001)
	Average	52,071 (86.7)	48,104 (86.7)	3967 (87.6)	
	Low	1443 (2.4)	1292 (2.3)	151 (3.3)	
Perceived health status	Healthy	43,300 (72.1)	40,045 (72.1)	3255 (71.9)	2.15 (0.340)
	Average	12,825 (21.4)	11,872 (21.4)	953 (21.1)	
	Not Healthy	3915 (6.5)	3597 (6.5)	318 (7.0)	
Academic performance level	High	23,420 (39.0)	21,721 (39.2)	1699 (37.5)	8.67 (0.010)
	Average	17,526 (29.2)	16,225 (29.2)	1301 (28.8)	
	Low	19,094 (31.8)	17,568 (31.6)	1526 (33.7)	

Data are presented as the number and percentage of respondents per variable. *p*-values were obtained from the χ^2^ test to investigate the difference between urban and rural. Total is the total number of participants who do not differentiate between urban and rural areas.

**Table 4 ijerph-18-02405-t004:** Comparison of health behaviors and health education among urban and rural adolescents.

Variables	Categories	Total (*n* = 60,040)	Urban (*n* = 55,514)	Rural (*n* = 4526)	χ^2^ (*p*)
		*n* (%)	*n* (%)	*n* (%)	
Breakfast	No	11,128 (18.5)	10,354 (18.7)	774 (17.1)	6.65 (0.010)
	Yes	48,912 (81.5)	45,160 (81.3)	3752 (82.9)	
Fast-food	No	11,676 (19.4)	10,682 (19.2)	994 (22.0)	19.76 (<0.001)
	Yes	48,364 (80.6)	44,832 (80.8)	3532 (78.0)	
Exercise (days per week)	Not Done	21,562 (35.9)	20,002 (36.0)	1560 (34.5)	19.88 (<0.001)
	1~3 days	25,976 (43.3)	24,069 (43.4)	1907 (42.1)	
	4 ≥ days	12,502 (20.8)	11,443 (20.6)	1059 (20.8)	
Sleep satisfaction	Enough	14,238 (23.7)	13,084 (23.6)	1154 (25.5)	12.78 (0.002)
	Average	20,098 (33.5)	18,561 (33.4)	1537 (34.0)	
	Not enough	25,704 (42.8)	23,869 (43.0)	1835 (40.5)	
Level of stress	A lot	24,312 (40.5)	22,419 (40.4)	1893 (41.8)	5.61 (0.050)
	Average	24,638 (41.0)	22,855 (41.2)	1783 (39.4)	
	None	11,090 (18.5)	10,240 (18.4)	850 (18.8)	
Current alcohol intake	No	15,030 (25.0)	13,762 (24.8)	1268 (28.0)	23.20 (<0.001)
	Yes	45,010 (75.0)	41,752 (75.2)	3258 (72.0)	
Current smoking	No	56,318 (93.8)	52,071 (93.8)	4247 (93.8)	0.01 (0.920)
	Yes	3722 (6.2)	3443 (6.2)	279 (6.2)	
Obesity	Underweight	3371 (5.6)	3118 (5.6)	253 (5.6)	21.33 (<0.001)
	Normal	48,060 (80.0)	44,569 (80.3)	3491 (77.1)	
	Overweight or obesity	8609 (14.2)	7827 (14.1)	782 (17.2)	
<Alcohol prevention education	No	34,214 (57.0)	31,645 (57.0)	2569 (56.8)	0.10 (0.751)
	Yes	25,826 (43.0)	23,869 (43.0)	1957 (43.2)	
Nutrition education	No	31,026 (51.7)	28,642 (51.6)	2384 (52.7	1.95 (0.162)
	Yes	29,014 (48.3)	26,872 (48.4)	2142 (47.3)	

Data are presented as the number and percentage of respondents per variable. *p*-values were obtained from the χ^2^ test to investigate the difference between urban and rural. Total is the total number of participants who do not differentiate between urban and rural areas.

**Table 5 ijerph-18-02405-t005:** Multinominal logistic regression of factors associated with body mass index (BMI) in urban and rural adolescents.

Variables	Urban (*n* = 55,514)										Rural (*n* = 4526)									
	Underweight					Overweight or Obesity					Underweight					Overweight or Obesity				
	B	S.E	*p*	OR	95% CI	B	S.E	*p*	OR	95% CI	B	S.E	*p*	OR	95% CI	B	S.E	*p*	OR	95% CI
School grade (Ref.: High school)	0.82	0.02	<0.001	2.27	2.17–2.38	−0.34	0.03	<0.001	0.71	0.68–0.75	0.86	0.09	<0.001	2.36	1.99–2.81	−0.17	0.09	0.052	0.85	0.71–1.00
Father’s education (Ref.: ≥University)	−0.04	0.03	0.127	0.96	0.91–1.01	0.15	0.03	<0.001	1.16	1.09–1.23	−0.05	0.10	0.581	0.95	0.78–1.15	0.18	0.10	0.093	1.20	0.98–1.47
Mother’s education (Ref.: ≥University)	−0.02	0.03	0.425	0.98	0.93–1.03	0.09	0.03	0.005	1.09	1.03–1.16	0.06	0.10	0.543	1.06	0.87–1.29	−0.09	0.10	0.412	0.92	0.75–1.13
Parents’ income level (Ref.: Middle & High)	−0.06	0.08	0.492	0.94	0.80–1.11	−0.24	0.08	0.002	0.79	0.68–0.92	−0.11	0.26	0.662	0.89	0.54–1.48	−0.51	0.21	0.011	0.60	0.40–0.90
Academic performance level (Ref.: Middle & High = 1)	−0.08	0.03	0.076	0.92	0.88–0.12	0.20	0.03	<0.001	1.22	1.16–1.29	−0.12	0.09	0.186	0.89	0.74–1.06	0.27	0.09	<0.001	1.32	1.11–1.56
Breakfast (Ref.: No)	−0.02	0.03	0.399	0.98	0.92–1.03	0.01	0.03	0.733	1.01	0.95–1.08	0.03	0.11	0.801	1.03	0.83–1.27	−0.06	0.11	0.621	0.95	0.76–1.18
Fast-food (Ref.: No)	−0.04	0.03	0.125	0.96	0.91–1.01	0.14	0.03	<0.001	1.15	1.08–1.22	0.01	0.10	0.894	1.01	0.84–1.23	0.14	0.10	0.173	1.15	0.94–1.40
Exercise (Ref.: No)	0.03	0.02	0.059	1.10	0.92–1.12	−0.20	0.03	<0.001	0.82	0.78–0.87	0.20	0.08	0.015	1.22	1.04–1.14	−0.27	0.09	<0.001	0.77	0.64–0.92
Current alcohol intake (Ref.: No)	−0.15	0.03	<0.001	0.86	0.81–0.90	0.09	0.03	0.001	1.10	1.04–1.16	−0.11	0.09	0.230	0.89	0.74–1.07	−0.01	0.09	0.951	0.99	0.83–1.19
Sleep satisfaction (Ref.: Enough & Average)	0.08	0.02	0.001	1.08	1.03–1.13	0.21	0.03	<0.001	1.23	1.17–1.30	0.07	0.09	0.436	1.07	0.90–1.26	0.22	0.09	0.013	1.25	1.05–1.49
Level of stress (Ref.: A Lot & Average)	−0.02	0.03	0.440	0.98	0.92–1.03	0.04	0.03	0.286	1.04	0.97–1.11	−0.07	0.10	0.478	0.93	0.76–1.14	0.10	0.11	0.361	1.11	0.89–1.38
Nagelkerke R^2^	0.053										0.053									

Dependent variable: BMI; BMI = weight (kg)/height (m^2^); Dependent variable reference group: Normal group, Underweight group: less than 5th percentile of BMI, Normal group: 5th percentile to less than the 85th percentile of BMI, Overweight or obesity group: 85th percentile or more of BMI; B: Estimate of the logistic regression coefficient; S.E: Standard Error of estimate; *p*: Significance probability; OR: Odds Ratio; 95%CI: 95% Confidence Interval.

## Data Availability

Not applicable.

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
