# Peer review of "Factors Affecting Obesity in Urban and Rural Adolescents: Demographic, Socioeconomic Characteristics, Health Behavior and Health Education"

_ijerph, 2021, doi:10.3390/ijerph18052405_

Round 1

Reviewer 1 Report

This paper identifies factors related to obesity in adolescents from urban and rural areas in Korea. While the authors demonstrate in the introduction to this paper that identification of these factors can lead to improvements in policy and programs focused on prevention and treatment of obesity, the discussion is in need of significant revision to identify how these results fit into the current literature in Korea and its clinical and policy implications. Additionally, there are a few things in the methods and results that need to be clarified.

Methods/ Results: For parent’s education, the categories are currently very confusing and do not appear to match the questionnaire. The questionnaire provides the options high school graduation or less and then university graduation or less, but the categories described in the methods and results say less than or equal to a high school education or greater than or equal to a university education. These two do not match up and thus this brings into question to validity of this measure.

Methods: Obesity measurement is indicated as a BMI≥ 25kg/m2. A BMI 25-29.9kg/m2 is considered overweight, and a BMI ≥30kg/m2 is indicative of obesity, thus this variable should be labeled as overweight/obesity. Please consider changing this throughout the paper.

Results section: The typed out results section should include numeric values as well. Currently it is all text. I recommend for the descriptions of Table 2 and Table 3, include the p-values of the variables discussed. For the logistic regression models, include the odds ratios and p values in parentheses.

Results: It is very interesting that perceived health status is not significantly different between urban and rural areas. Please comment on this in the discussion and what you think it means.

Discussion:

The discussion lacks a clear structure and presentation of arguments. Several paragraphs jump around between variables and topics without presenting a clear argument or conclusion. For example, in the fifth paragraph, the last sentence makes a conclusion related to physical activity, but the preceeding sentences are about drinking and smoking rates.

Throughout the discussion there is a general sentiment that the implications of these results are to create health education and promotion policies. Yet, health education was not significantly associated with overweight/obesity in this study. Please be more specific about what types of health education and promotion policies may be needed. Also consider obesity prevention and treatment methods focused solely on health education have not been successful.

Reviewer 2 Report

Major concern: Do you consider sample weight for regression analysis? As the analysis is based on survey data, it should be provided by the data collection authorities. Use appropriate sample weight (for individual).

Self-reported BMI is a major concern. So, you can include "self-reported" in title such as: "Factors Affecting Self-reported Obesity in Urban and Rural ...."

Minor concern: Include a pooled data regression (including rural and urban) so that readers can understand the national/total picture of the country. It is not necessary to discuss in the text.

Reviewer 3 Report

1- In Materials and Methods: Obesity is classified according to adults' BMI classification. Can the authors comment on the fact that the majority of the sample was adolescents, and their BMI has to be classified according to the growth charts?
Note: "WHO defines 'Adolescents' as individuals in the 10-19 years age group and 'Youth' as the 15-24 year age group."

2- In Materials and Methods: overweight was not considered and classified as obesity. Can the authors comment on this or list it as a limitation?

3- Can the authors expand on the fact that the survey was self-reported? It seems some of the questions had "YES" or "NO" answers only, while these questions are supposed to have at least three answers. For instance, questions about breakfast, fast food, and obesity would benefit from more details. Also, sleep satisfaction and level of stress had to be more objective. i.e., sleep hours or sleep quality (can not be usually measured by 1 question). Perhaps the authors might consider expanding on this in the discussion or acknowledging these details in the limitations.

Reviewer 4 Report

This is a cross-sectional study examining the associations between socioeconomic characteristics, health behavior, health education experience, and weight status in Korean adolescents living in urban and rural areas. This study uses the data of the 14th Youth Health Behavior Survey conducted in Korea (2018). It assesses a Korean sample of 60040 Korean adolescents. This area of inquiry is one that I very much hope the authors will pursue in future research. Although the admirable effort of the authors to explain the theoretical background of this study. I think that the authors could take a step further in making clear the novelty/innovation of this study, the relevance of the statistical analysis, as organizing study methodology and results in a more intelligible and explanatory way. Editing by a native speaker is required throughout the manuscript. Finally, I believe that this data has a great potential for more robust and innovative statistical analysis.

General Comments

Abstract

  1. It will be important to reformulate the abstract in order to include information about, the mean age of the participants and gender frequencies.

Introduction

  1. The pertinence and innovation of this study are not clear. It will be helpful to interpret data in order to make a stronger case for the pertinence of this work. 
  2. I would suggest that authors reformulate this section integrating in a logical way the various ideas/paragraphs and topics addressed in the introduction.
  3. Sometimes the writing lacks a scientific tone and, although I am not a native English speaker, I experienced some trouble in understanding the study aims and results at a first glance.
  4. “Poverty and health issues among adolescents create a vicious cycle, leading to poor health levels, health inequity, low income, poor education, and difficultly getting employment [1].” - Please support this sentence with the appropriate references (just one reference to support this general idea seems insufficient).
  5. “In Korea in the 2000s, the issue of health inequality between the rich and the poor was brought to light, causing an increase in research on the subject [7]. “ - Please support this sentence with the appropriate references (just one reference to support this idea of increased research seems insufficient).
  6. “Despite the need for early intervention of health inequality in adolescence, there is not much research on health inequality among the adolescent population or policies to alleviate this inequality [9]. It is particularly necessary to examine the factors of health inequality caused by regional differences (i.e., urban and rural areas).” This idea seems disconnected from the previous paragraph.
  7. “One school doctor, one school pharmacist, and one health teacher shall be assigned to elementary schools with 18 classes or more, on the other hand, one school doctor, one school pharmacist, one health teacher could be assigned to elementary schools with less_than 18 classes” Can please reformulate this sentence to make it easier to understand?
  8. “If there are differences in health education between regions, adolescents residing in certain areas may lack knowledge about attitudes and practices that affect health, which can lead to increased obesity rates due to nutritional imbalance and lack of exercise. “– Obesity is a multifactorial disease. Nutritional imbalance and lack of exercise are just two factors from the many factors related to obesity.
  9. “Based on previous studies, we concluded that the biological, genetic, and socioeconomic characteristics of individuals are important variables that can cause health problems, such as obesity, and that the local community environment has a direct effect on health [13].” – Authors refer “previous studies” but just one reference to support this sentence. Please support this sentence with the appropriate references. I also suggest to avoid the expression “we concluded”.
  10. “Therefore, it is necessary to understand the factors that affect obesity in adolescents in urban and rural areas as a major indicator of their health inequality [15]. Health behavior is very meaningful in that there is a possibility of improvement compared to other health inequality factors such as socioeconomic status and parents' income level [16]. Therefore, research on basic data identifying regional differences and health disparities by examining the health behavior, health education experience, and degree of obesity of urban and rural adolescents is needed to establish a more comprehensive health promotion policy.”- This paragraph has important and interesting information, nevertheless English editing is needed.
  11. At study aims what do you mean by “and level of obesity”?

Methods

  1. The statistical analysis section is very limited. I would suggest a more detailed section explaining how you handled missing data and performed the statistical analysis. It will also be important to know if the variables under study were normally distributed. Did you have missing data? If yes how did you handled it? What assumptions for binary logic regression did the authors consider?  The statistical analysis described in table 1 were not included in this section.

  1. It is important to clarify throughout the paper if you are referring to the biological sex of participants or to their self-attributed/chosen “gender” when using the word “gender” rather than sex.

  1. “This cross-sectional, descriptive study analyzed secondary data from the 14th Youth 105 Health Behavior Survey conducted nationwide in Korea in 2018”- What do you mean by secondary data?

  1. “Students with long-term absences, disabilities, and text reading disabilities were excluded”.- Please explore in the discussion section how these exclusion criteria can create bias in the interpretation of your study results.

  1. Table legends and information about the statistical tests performed are missing in Tables 1, 2, and 3. Table 1 should be in the results section. Please indicate in the text or in table 1 the minimum and maximum range for age.

  1. Those with a BMI of less than 25 kg/m2 were classified as non-obesity, and those with a BMI of 25 kg/m2 or more were classified as obesity.”- This is a major limitation of this study. In growing children/adolescents BMI varies with age and sex, thus the best practices with pediatric samples recommend authors to only report and use in statistical analysis BMI z-score. Otherwise compare BMI values between gender etc. without standardization is not feasible due to the different development paths in this age group. Besides, I would recommend using the standard (WHO, CDC, Etc.) or Korean BMI cutoffs for children/adolescents. At last, I would encourage authors also to explore differences between normal weight, overweight, and obesity (for example “Normal weight (BMI <25), Overweight (BMI ≥25) and Obesity (BMI ≥ 30)) following BMI z score cutoffs if possible. 

Results

  1. “No significant differences regarding health education, alcohol prevention education, or nutrition education were reported.” – The values of the statistical test performed should be reported in the text or in the table.
  2. It will be valuable to reformulate the study results in accordance with the different overweight (BMI >25 <30)/obesity (BMI >30) classification status rather than using “obesity” as the same as BMI > 25. The reference to the weight status classification (WHO, CDC, etc.) should be stated in the text (methods section).
  3. Additional statistical analysis could be performed to enrich the paper such as correlations or a path analysis (with a multigroup analysis with rural/urban as the study groups) to understand more clearly the relationships between the variables under study for each group (rural vs urban adolescents). Including some illustrations (graphs) describing health behaviors in rural and urban adolescents will be appreciated.

Discussion

  1. Study limitations should be detailed before the Conclusion section.
  2. In your sample there is a considerable discrepancy in the sample size of rural and urban adolescents and BMI was evaluated by self-report. These are important study limitations and should be identified in the limitations paragraph.

Thank you for the opportunity to revise your valuable work. I hope that my comments help to improve your paper.

Round 2

Reviewer 2 Report

Ok.

Author Response

Reviewer 2's comment is written as'OK', so there are no modifications related to this. However, we have once again reviewed and corrected the content of the text as a whole.

Thank you.

Reviewer 4 Report

I found the authors to be responsive in their revisions to the manuscript titled " Factors Affecting Obesity based on Self-reported in Urban and Rural Adolescents: Socio-economic Characteristics, Health Behavior, and Health Education." Specifically, I believe that the introduction is now improved and provides better context for the main points of the study. The additional clarifications surrounding the sample and methods are also helpful. The following are suggestions for further improvement of the manuscript.

Title:

1. Please consider revising the title. The word “self-reported” seems misplaced. “Factors Affecting Obesity based on Self-reported in Urban and Rural Adolescents: Socio-economic Characteristics, Health Behavior, and Health Education”

Introduction:

1. Please consider revising the following sentence. I had difficulty in understanding its meaning “Previous studies that measured the health gap between regions or classes at the national level received attention [10, 11]. Such findings, Health inequality existed between the metropolitan and non-metropolitan areas, as well as between rural and urban areas, and local health conditions were closely related to regional deprivation [11].".

2. Regarding aims, the term “general characteristics” can be unclear. Also please consider substituting “differences in the degree of obesity in urban and rural adolescents” by “ differences between urban and rural adolescents in weight status.”.

Methods

1. I noticed that the authors stated that normal distribution was assumed. In this scope, the Spearman correlations are not appropriated. In addition, table 2 should identify in legends the type of statistical analysis performed.

Results

  1. The weight status classification approach used in this study still raises some scientific concerns. In the methods section is not clear if the BMI percentiles were used to classify each participant or if the authors are just giving information regarding BMI classification. In addition, aggregating underweight and normal-weight participants in the same group can raise methodologic and statistical concerns, since underweight participants can have specific sociodemographic/clinical characteristics that prevent them to be in the same group of normal-weight participants. I would suggest considering three weight status groups in your statistical analysis: underweight/normal weight/overweight-obesity.

    Thank you for the opportunity to revise your valuable work. I hope that my comments help to improve your paper.
